# Artificial intelligence augmented home sleep apnea testing device study (AISAP study)

**Sunil Sharma**[1]*, **Kassandra Olgers**[1], **Scott Knollinger**[2], **Saivenkat Somisetty**[3], **Calvin Seol**[4], **Naveena Yanamala**[5]

1 Division of Pulmonary, Critical Care and Sleep Medicine, West Virginia University, Morgantown, WV, United States of America, 2 Department of Respiratory Care, Ruby Memorial Hospital, Morgantown, WV, United States of America, 3 Montgomery Township High School, Skillman, NJ, United States of America, 4 Eberly College of Arts and Science, West Virginia University, Morgantown, WV, United States of America, 5 Rutgers Robert Wood Johnson Medical School, Division of Cardiovascular Disease and Hypertension, New Brunswick, NJ, United States of America

* sunil.sharma@hsc.wvu.edu

## Abstract

### Study objective

This study aimed to prospectively validate the performance of an artificially augmented home sleep apnea testing device (WVU-device) and its patented technology.

### Methodology

The WVU-device, utilizing patent pending (US 20210001122A) technology and an algorithm derived from cardio-pulmonary physiological parameters, comorbidities, and anthropological information was prospectively compared with a commercially available and Center for Medicare and Medicaid Services (CMS) approved home sleep apnea testing (HSAT) device. The WVU-device and the HSAT device were applied on separate hands of the patient during a single night study. The oxygen desaturation index (ODI) obtained from the WVU-device was compared to the respiratory event index (REI) derived from the HSAT device.

### Results

A total of 78 consecutive patients were included in the prospective study. Of the 78 patients, 38 (48%) were women and 9 (12%) had a Fitzpatrick score of 3 or higher. The ODI obtained from the WVU-device corelated well with the HSAT device, and no significant bias was observed in the Bland-Altman curve. The accuracy for ODI > = 5 and REI > = 5 was 87%, for ODI> = 15 and REI > = 15 was 89% and for ODI> = 30 and REI of > = 30 was 95%. The sensitivity and specificity for these ODI /REI cut-offs were 0.92 and 0.78, 0.91 and 0.86, and 0.94 and 0.95, respectively.

### Conclusion

The WVU-device demonstrated good accuracy in predicting REI when compared to an approved HSAT device, even in patients with darker skin tones.

**Data Availability Statement:** All relevant data are within the paper and its Supporting Information files.

**Funding:** This work is supported in part by funds from the National Science Foundation (NSF: Award # 2125872) and the Department of Medicine, West Virginia University, Morgantown, WV The funders had no role in study design, data collection and analysis, decision to publish, or preparation of the manuscript.

**Competing interests:** Sunil Sharma, M.D. is on the speaker bureau for Zoll Respicardia Inc No other COI to report

## Introduction

Obstructive sleep apnea is a highly prevalent and underdiagnosed condition. Recent estimates of its prevalence suggest over a billion people in the world may be suffering from it [1]. Undiagnosed and untreated sleep apnea is associated with significant cardiovascular morbidity including, hypertension, congestive heart failure (CHF), atrial fibrillation, stroke, and acute coronary syndrome (ACS) [2–4]. Simple, accurate, and reproducible screening and testing at home can play a crucial role in the early detection of this potentially fatal disease. While a complete polysomnography (PSG) recording remains the gold standard for diagnosing Obstructive Sleep Apnea (OSA), its reliance on laboratory facilities and trained technicians for manual setup and scoring renders it both expensive and time-consuming. Home sleep apnea testing (HSAT) on the other hand has high failure rates, cumbersome for patients and only captures single night data only [5]. As a result, there is a demand for innovative approaches that can facilitate the automatic diagnosis and evaluation of Obstructive Sleep Apnea (OSA) severity. Additionally, application of Artificial Intelligence (AI) has shown promise in predicting, diagnosing, scoring and classifying sleep apnea accurately and cost effective way [6].

The WVU Device Platform has three main parts: a finger-tip pulse oximeter that securely fits on your index finger, a charging cable, and patented AI-augmented software for biomarker discovery and prediction of OSA severity. The finger-tip sensor is a two-channel device and is designed to acquire pulse oximetry and PPG. signals from the proximal phalangeal radialis indicis artery of the index finger (Fig 1). It provides a user-friendly interface, simpler than the 4-channel devices commonly used in hospital settings. The AI-augmented algorithm, integrating time-series analysis with regression/classification approaches for preprocessing SPO2 data to detect ODI and other biomarkers and predict the severity of OSA, underwent training on our prior data from 365 patients in overnight sleep studies using the 4% desaturation hypopnea criteria [7]. The WVU device utilizes a combination of oxygen saturation changes and additional clinical information (e.g., demographics, comorbidities, risk factors) to identify desaturation events and automatically diagnose and assess the severity of OSA (Fig 1). This study evaluates the WVU device in consecutive hospitalized patients with significant comorbidities with simultaneous use of a 4-channel CMS-approved HSAT device.

## Methodology

This prospective study was conducted at a tertiary care university hospital between November 2022 and June 2023. The study was approved by the West Virginia University Institutional Review Board (WVU IRB) (IRB # 2205569959A001). The University Hospital has a formal sleep apnea screening program under which patients admitted to medical wards who have

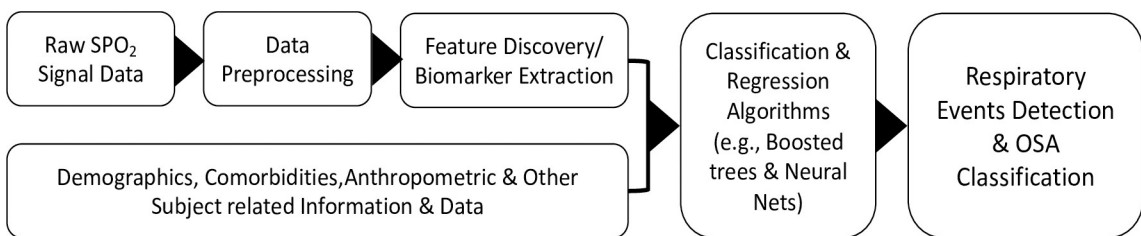

**Fig 1. Schematic overview of the process for classifying sleep apnea severity and detecting respiratory events using machine learning algorithms.** This figure provides a detailed view of the workflow used to diagnose sleep apnea and detect specific respiratory events through machine learning techniques. The process encompasses two primary phases, executed in a structured sequence: initially, the detection of specific respiratory events, followed by the classification of sleep apnea severity. Both phases employ a combination of classification and regression-based algorithms to ensure a comprehensive and accurate analysis.

BMI $\geq$ 30 and STOP positive are offered a home sleep apnea test (HSAT) to screen for undiagnosed sleep apnea [8]. The HSAT device is a commercially available Type 3 device with 4 channels, including airflow, chest movement, pulse oximetry, and heart rate. The screening is performed once the patient has recovered from an acute condition and is on minimal or no oxygen support. Additional inclusion criteria for final analysis included a minimum data recording time of 2 hours or more for both the test (WVU device) and control device (HSAT). Consecutive adult patients admitted to the medical ward who were clinically undergoing an HSAT were approached and consented to simultaneously test with the WVU-device after informed consent was obtained.

The device was placed in the contra-lateral hand by a trained respiratory therapist. The device was retrieved by the respiratory therapist the following morning and data was downloaded. A board-certified sleep physician-reviewed and interpreted the data. The staff were blinded to the results of HSAT and WVU-device which were analyzed by a non-clinical scientist. The WVU device was calibrated to identify events with a 4% drop in oxygen. A Fitzpatrick classification scale [9] was used to accurately assess and include a proportion of patients with darker skin tones (IV-VI) as suggested by the FDA as the device was pulse-oximeter based [10].

## Statistical analysis

Statistical analysis was performed using R (ref) to assess the accuracy of the WVU-device AI-enabled Platform in predicting respiratory events. Continuous variables are expressed as the mean ± standard deviation, and categorical variables are expressed as percentages. To assess the agreement between the WVU-device and the CMS-approved HSAT device, we performed a Pearson correlation coefficient, and Bland-Altman analysis. The significance threshold was set at a two-sided p-value <0.05. Finally, the Receiver Operating Characteristic (AUC-ROC) analysis was conducted to further assess the accuracy, precision, and reliability of the two measurements (i.e., WVU-device ODI and HSAT REI values) above specific clinically relevant thresholds (e.g., ODI $\geq$ 5, REI $\geq$ 5; AHI $\geq$ 15, REI $\geq$ 15; ODI $\geq$ 30, REI $\geq$ 30). The Area Under the ROC Curve (AUC) was calculated to quantify the overall discriminative ability of the WVU-device in predicting the sleep apnea severity thresholds. An AUC of 1 indicates perfect discrimination, while 0.5 suggests no discrimination beyond chance.

## Results

A total of seventy-eight consecutive adult patients (age >18 years) admitted to the medical wards who met the inclusion criteria were studied and analyzed. Of the 78 patients, 40(52%) were males, with a mean BMI of 36.82 ± 8.22 and a mean neck circumference of 19 ± 1.41 inch. Of the 78 patients, 68 (87%) were Caucasians, 7 (9.1%) were African Americans. Of the total cohort 59(77%) had hypertension, 23(30%) had congestive heart failure (CHF), 27(35%) had type 2 diabetes mellitus, 22(29%) had coronary artery disease (CAD) and 17(22%) had chronic obstructive pulmonary disease (COPD). Of the total of 78 patients 51(66%) were classified as obstructive sleep apnea (OSA) with AHI of $\geq$ 5 and 16 (21%) were classified as severe OSA with AHI $\geq$ 30. A total of 9 patients (12%) had a darker skin tone (Fitzpatrick class IV to VI)

Sleep parameters revealed a mean HSAT AHI of 17.9 ± 20.15 and mean ODI derived from WVU-device of 19.83 ± 22.06 (Table 1).

A strong and statistically significant correlation was observed between the ODI derived from the WVU device and the AHI measured by the 4-channel HSAT device (r = 0.885, p<0.001; Fig 2). This demonstrates a robust positive relationship between the two variables, reinforcing the reliability and validity of the relationship between the two measures.

**Table 1. Baseline characteristics, sleep parameters and comorbidities for the group (N = 78).**

| Demographics | |
|---|---|
| Age (years) | 57.1 ± 14.6 |
| Males | 40 (51%) |
| BMI | 40.9 ± 13.8 |
| Neck Circumference(inch) | 18.1 ± 2.2 |
| Race (White) | 68 (87%) |
| Race (African American) | 7 (10.3%) |
| Race (Asian) | 2 (2.6%) |
| Fitzpatrick IV-VI | 9 (12%) |
| Sleep Study Parameters | |
| AHI | 17.9 ± 20.2 |
| HSAT ODI | 20.3 ± 19.8 |
| WVU-device ODI | 20.1 ± 22.1 |
| AHI<5 | 27 (34%) |
| AHI> = 5 & AHI<15 | 17 (22%) |
| AHI> = 15 & AHI<30 | 18 (23%) |
| AHI> = 30 | 16 (21%) |
| Comorbidities | |
| HTN | 59 (76%) |
| CHF | 23 (30%) |
| DM | 27 (35%) |
| CAD | 22 (28%) |
| COPD | 17 (22%) |

The Bland-Altman plot illustrating the difference between AHI and WVU-device ODI did not reveal any significant bias and showed a mean difference of -2.06 events/h between AHI and WVU-device ODI was −2.3 (Fig 3). The accuracy for ODI > = 5 and AHI > = 5 was 87%, for ODI> = 15 and AHI > = 15 was 89% and for ODI> = 30 and AHI of > = 30 was 95% (Fig 4). The sensitivity and specificity for these ODI/AHI cut-offs were 0.92 and 0.78, 0.91 and 0.86 and 0.94 and 0.95. The positive predictive values for these cut-offs were 0.89, 0.84, 0.83 and negative predictive values were 0.84, 0.93, and 0.98 (Table 2).

A subset analysis of patients with a Fitzpatrick score of 3 or more (n = 9) revealed an accuracy of 0.89, sensitivity of 1.00, specificity of 0.8, positive predictive value of 0.8, and negative predictive value of 1.00.

## Discussion

Our findings reveal a strong correlation between the ODI derived from the WVU device and the AHI measured by the 4-channel HSAT device. The negative predictive values for clinically significant sleep apnea, defined as an AHI / ODI of 15 and 30 events per hour, were notably high at 93% and 98%, indicating a low likelihood of missing the disease.

Currently, available HSAT devices are often cumbersome to use due to their requirement for multiple channels (four or more) and have been shown to have a high failure rate of nearly 33% [5]. This highlights the need for a more user-friendly screening tool. Given the high prevalence of sleep apnea and its underdiagnosis, the WVU device could hold promise as an ideal tool for detecting sleep apnea in both hospitalized and ambulatory settings. Emerging data also suggest that screening for sleep apnea in hospitalized patients and early initiation of positive airway pressure therapy may lead to reduced readmissions and healthcare costs [11–13].

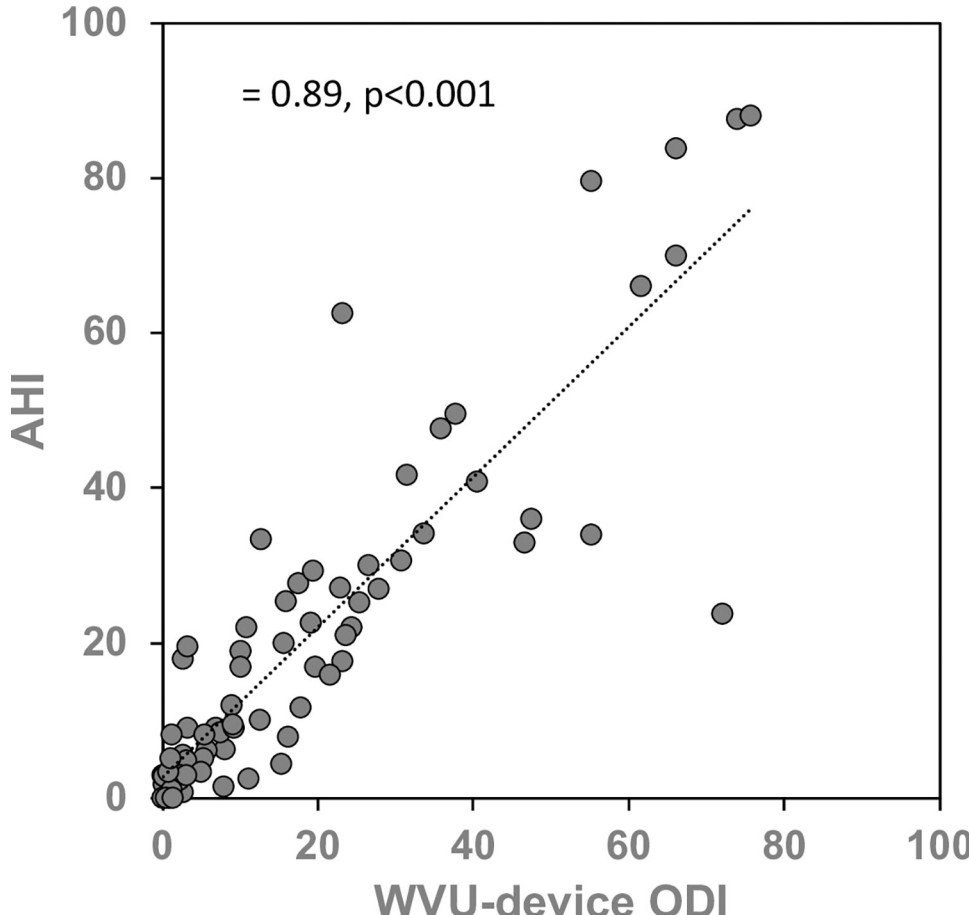

**Fig 2. Scatterplot comparing HSAT device AHI to WVU-device ODI.** AHI = apnea-hypopnea index, ODI = oxygen desaturation index.

There is a significant concern within the scientific community regarding the reliability of pulse oximeters in patients with darker skin tones [14]. Studies have consistently shown that the oximeters tend to overestimate oxygen saturations in individuals with darker skin tones [14]. However, the oxygen signal sampling and the proprietary algorithm utilized by the WVU device appear to overcome this barrier. Notably, our study is the first to include patients with a Fitzpatrick classification scale of IV or higher (12%) to specifically assess darker skin tones. An additional advantage of the WVU device is its multi-night data capability, which recent studies have found to be more accurate [15]. Compared to prior studies with high-resolution pulse-oximetry, the WVU-device demonstrates an overall improvement in PPV (89% vs 82%, 84% vs 76%, and 83% vs 73% for REI≥5, ≥15, and ≥30 respectively), specificity(78% vs 67%, 89% vs 84%, and 95% vs 93% for REI≥5, ≥15, and ≥30 respectively) and accuracy (87% vs 84%, 89% vs 87%, and 95% vs 91% for REI≥5, ≥15, and ≥30 respectively), indicating enhanced precision in its diagnostic capabilities [7]. Moreover, WVU-device's algorithm also incorporates a robust self-learning ability, enabling continuous refinement and optimization of diagnostic performance. This self-learning capability holds the promise of enhancing precision, particularly for racial minorities, thus addressing healthcare disparities. AI augmented WVU device can also help in development of clinical prediction models that rely on accurate oxygen desaturation index (ODI) especially in patients planning to have surgical procedures such as

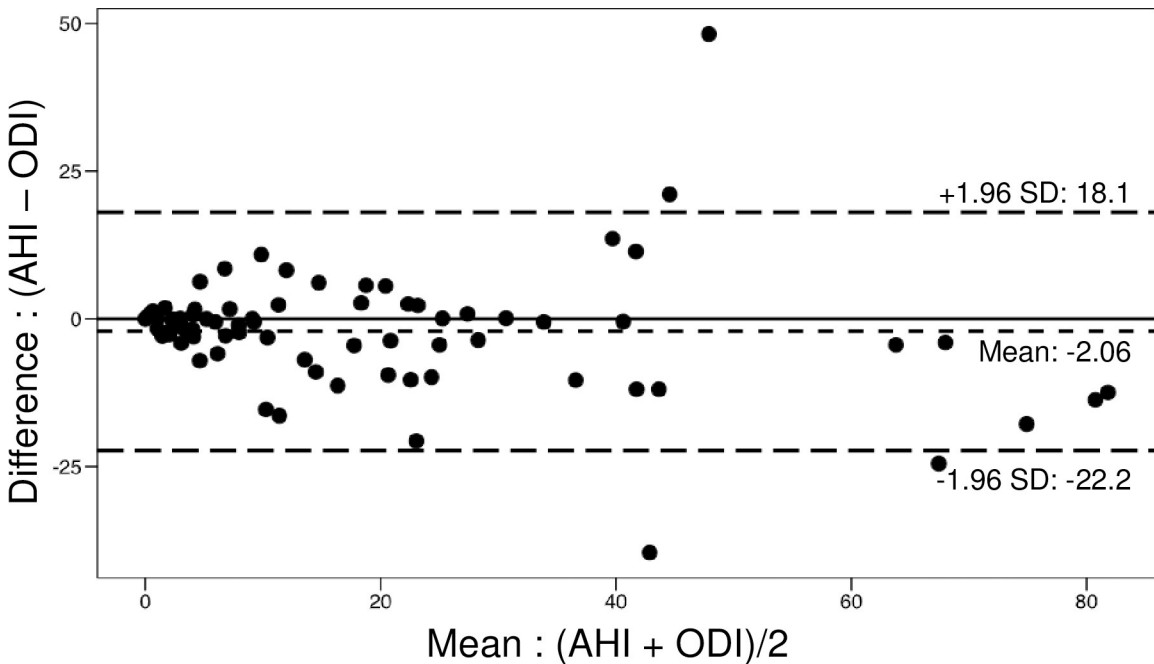

**Fig 3. The Bland-Altman plot comparing the Apnea-Hypopnea Index (AHI) from the HSAT device with the Oxygen Desaturation Index (ODI) from the WVU device.**

CABG, bariatric surgery or arthroplasty among others. The device may help with better triaging of patients and reducing wait time for surgeries [16].

The limitations of the study include that despite its multi-night capability the device was analyzed for one night only. This was because these patients were undergoing a clinically indicated one-night HSAT study during their hospital stay and a multi-night study would have inadvertently extended their hospital stay. We performed the sleep studies in a hospitalized environment, although these studies were unattended to replicate conditions at home. This proof-of-concept study would have to be validated in a home environment in a multi-night

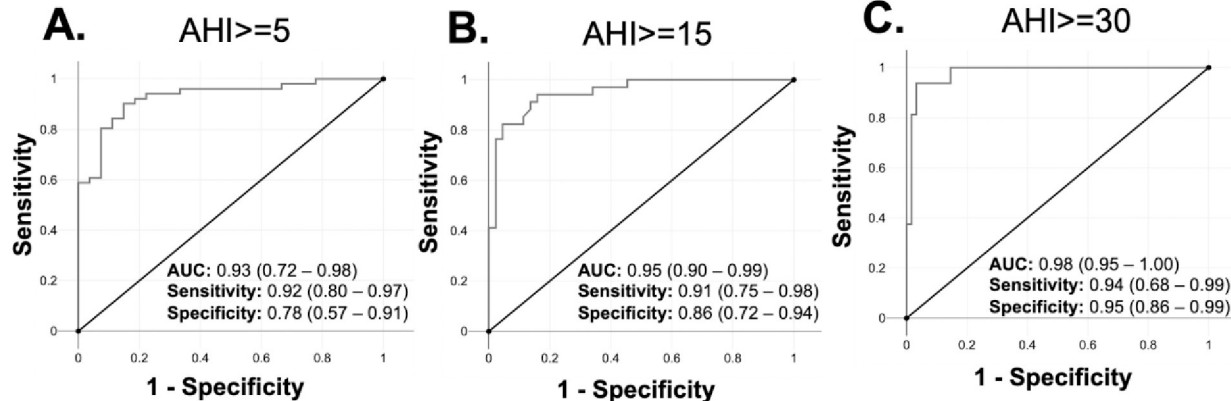

**Fig 4. Receiver operating characteristic curve (ROC) of WVU-device ODI for prediction of the HSAT-device derived AHI thresholds.** This figure displays ROC curves for the WVU-device Oxygen Desaturation Index (ODI) predicting Apnea-Hypopnea Index (AHI) thresholds of (A) 5, (B) 15, and (C) 30 events per hour. Each curve shows the balance between sensitivity and specificity for different thresholds. AUC values indicate the predictive accuracy of WVU-device ODI for each AHI threshold. Performance metrics including sensitivity, specificity and AUC for each AHI threshold, derived from ROC analysis on a dataset of n = 78 samples is also provided.

**Table 2. Performance metrics of WVU-device ODI at HSAT device AHI thresholds of 5, 15, and 30 events/h.** AUROC = area under the receiver operator curve, NPV = negative predictive value, PPV = positive predictive value.

| Sample Size (n = 78) | AUROC | Accuracy | Sensitivity | Specificity | PPV | NPV |
|---|---|---|---|---|---|---|
| AHI> = 5, ODI> = 5 | 0.93 | 0.87 | 0.92 | 0.78 | 0.89 | 0.84 |
| AHI> = 15, ODI> = 15 | 0.95 | 0.89 | 0.91 | 0.86 | 0.84 | 0.93 |
| AHI> = 30, ODI> = 30 | 0.98 | 0.95 | 0.94 | 0.95 | 0.83 | 0.98 |

protocol. Secondly, as with most hospitalized patients, our cohort had significant comorbid conditions. Testing the community with a lesser burden of comorbid conditions may be required to establish generalizability. Furthermore, machine learning algorithms have their own limitations including need for external validation on external data and potential for bias if the training data is not representative of the patient population to be studied. The latter has the risk of widening health disparities in ethnic minorities along with introducing regional bias.

Potential challenges to incorporate WVU device could be limited comfort levels of physicians with this technology and inertia of already established care pathways. Future research should focus on validating the device in a larger, more diverse population, along with evaluating its impact on patient outcomes.

## Conclusion

WVU device is a simple, cost-effective screening tool that correlates well with approved, commercially available level 3 HSAT devices. Considering that a large proportion of patients with sleep apnea remain undetected, this tool may be beneficial to healthcare organizations to evaluate patients with a high risk of sleep apnea, develop clinical prediction models and help reduce healthcare disparities.

## Supporting information

**S1 Dataset.**
(XLSX)

## Author Contributions

**Conceptualization:** Sunil Sharma, Kassandra Olgers, Scott Knollinger, Saivenkat Somisetty, Calvin Seol, Naveena Yanamala.

**Data curation:** Sunil Sharma, Kassandra Olgers, Scott Knollinger, Saivenkat Somisetty, Calvin Seol, Naveena Yanamala.

**Formal analysis:** Sunil Sharma, Naveena Yanamala.

**Funding acquisition:** Sunil Sharma.

**Investigation:** Sunil Sharma, Naveena Yanamala.

**Methodology:** Sunil Sharma, Kassandra Olgers, Scott Knollinger, Saivenkat Somisetty, Calvin Seol, Naveena Yanamala.

**Project administration:** Sunil Sharma, Scott Knollinger, Calvin Seol, Naveena Yanamala.

**Resources:** Sunil Sharma, Naveena Yanamala.

**Software:** Sunil Sharma, Saivenkat Somisetty, Calvin Seol, Naveena Yanamala.

**Supervision:** Sunil Sharma, Naveena Yanamala.

**Validation:** Sunil Sharma, Naveena Yanamala.

**Visualization:** Sunil Sharma, Kassandra Olgers, Scott Knollinger, Naveena Yanamala.

**Writing – original draft:** Sunil Sharma, Kassandra Olgers, Scott Knollinger, Naveena Yanamala.

**Writing – review & editing:** Sunil Sharma, Kassandra Olgers, Scott Knollinger, Saivenkat Somisetty, Calvin Seol, Naveena Yanamala.

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
