## [Decision Letter · Decision Letter 0]

1 Apr 2024

PONE-D-24-04849Artificial Intelligence Augmented Home Sleep Apnea Testing Device Study (AISAP study)PLOS ONE

Dear Dr. Sharma,

Thank you for submitting your manuscript to PLOS ONE. After careful consideration, we feel that it has merit but does not fully meet PLOS ONE’s publication criteria as it currently stands. Therefore, we invite you to submit a revised version of the manuscript that addresses the points raised during the review process.

**ACADEMIC EDITOR: ** Hard Recommendations:Please mention name of the device if possible.The introduction should be made stronger by mentioning recent research that AI can fill the gap of underdiagnosis of sleep disorders. The discussion and conclusion can be made stronger by mentioning that this AI augmented WVU device can help in development of clinical prediction models that rely on accurate oxygen desaturation index (ODI) especially in cardiac patients planning to have cardiac procedures such as CABG or other procedure. SUGGESTIONS:The discussion section could further explore how the WVU device might be integrated into clinical care pathways for sleep apnea. What are the potential benefits and challenges of implementing this device in practice?Commenting on cost-effectiveness: If data are available, the authors could discuss the potential cost-effectiveness of the WVU device compared to traditional HSAT. This is an important consideration for adoption of new technology.Expanding the limitations section: The authors could mention any potential limitations related to the AI algorithm, such as the need for external validation on other datasets or the potential for bias if the training data were not representative.Discussing future directionsEnhancing figures: While the figures are clear, some additional annotations or labels could help guide the reader's interpretation.==============================

We look forward to receiving your revised manuscript.

Kind regards,

Harpreet Singh Grewal

Academic Editor

PLOS ONE

“This work is supported in part by funds from the National Science Foundation (NSF: Award # 2125872)”

3. In the online submission form, you indicated that your data will be submitted to a repository upon acceptance.  We strongly recommend all authors deposit their data before acceptance, as the process can be lengthy and hold up publication timelines. Please note that, though access restrictions are acceptable now, your entire minimal  dataset will need to be made freely accessible if your manuscript is accepted for publication. This policy applies to all data except where public deposition would breach compliance with the protocol approved by your research ethics board. If you are unable to adhere to our open data policy, please kindly revise your statement to explain your reasoning and we will seek the editor's input on an exemption.

Reviewers' comments:

Reviewer's Responses to Questions

**Comments to the Author**

1. Is the manuscript technically sound, and do the data support the conclusions?

Reviewer #1: Yes

Reviewer #2: Yes

2. Has the statistical analysis been performed appropriately and rigorously? 

Reviewer #1: Yes

Reviewer #2: Yes

3. Have the authors made all data underlying the findings in their manuscript fully available?

Reviewer #1: No

Reviewer #2: Yes

4. Is the manuscript presented in an intelligible fashion and written in standard English?

Reviewer #1: Yes

Reviewer #2: Yes

5. Review Comments to the Author

Reviewer #1: Data Access: the authors have not made all data fully available without restriction. In the Data Availability statement, the authors state:

"Data are available from the WVU Institutional Data Access / Ethics Committee (contact via email) for researchers who meet the criteria for access to confidential data"

This indicates there are some restrictions on openly sharing the underlying data. Researchers can request access through the WVU Institutional Ethics Committee, but would need to meet certain criteria to obtain the confidential data.

Minor Refinements:

Expanding the introduction: The authors could provide a bit more context on the current limitations of home sleep apnea testing (HSAT) devices and the potential advantages of the WVU device. This would help highlight the significance of their work.

Providing more details on the WVU device: The methods section could include some additional technical specifications about the WVU device, such as the specific sensors used, sampling rates, or key components of the AI algorithm. This would allow readers to better understand and potentially replicate the technology.

Reporting confidence intervals: In addition to the point estimates for accuracy, sensitivity, specificity, etc., the authors could report 95% confidence intervals. This would provide a measure of the precision of these estimates.

Discussing implications for clinical practice: The discussion section could further explore how the WVU device might be integrated into clinical care pathways for sleep apnea. What are the potential benefits and challenges of implementing this device in practice?

Commenting on cost-effectiveness: If data are available, the authors could discuss the potential cost-effectiveness of the WVU device compared to traditional HSAT. This is an important consideration for adoption of new technology.

Expanding the limitations section: The authors could mention any potential limitations related to the AI algorithm, such as the need for external validation on other datasets or the potential for bias if the training data were not representative.

Discussing future directions: The conclusion could outline some key next steps for research, such as validating the device in a larger, more diverse population or conducting studies on the impact of the device on patient outcomes.

Enhancing figures: While the figures are clear, some additional annotations or labels could help guide the reader's interpretation. For example, labeling the sensitivity and specificity directly on the ROC curves.

Reviewer #2: Dear authors, thanks for bringing this research to show the importance of new device capable of measuring oxygen level in people with different skin colors with fair and comparable accuracy with FDA approved device. It would be great to mention the name and model of FDA approved device. The introduction can be made stronger by mentioning recent research that AI can fill the gap of underdiagnosis of sleep disorders. The discussion and conclusion can be made stronger by mentioning that this AI augmented WVU device can help in development of clinical prediction models that rely on accurate oxygen desaturation index (ODI) especially in cardiac patients planning to have cardiac procedures such as CABG or other procedure. There are minor typo errors such as full stop after PPG at one place. The date format in methods can be modified to US standard. Otherwise, well written article!

6. PLOS authors have the option to publish the peer review history of their article (what does this mean?). If published, this will include your full peer review and any attached files.

Reviewer #1: **Yes: **Ankit Virmani

Reviewer #2: **Yes: **Ram K Verma

---

## [Author Response · Author response to Decision Letter 0]

10 Apr 2024

Hard Recommendations:

• Please mention name of the device if possible.

Response: The device has not yet been commercialized and hence we do not have a confirmed name and since it is patented to West Virginia University we have named it WVU device for this scientific paper.

• The introduction should be made stronger by mentioning recent research that AI can fill the gap of underdiagnosis of sleep disorders. 

Response: We have added the recent paper by Bazoukis etal on AI to fill the gap of understanding underdiagnosis of sleep apnea in the introduction section.

Bazoukis G, Bollepalli SC, Chung CT, Li X, Tse G, Bartley BL, Batool-Anwar S, Quan SF, Armoundas AA. Application of artificial intelligence in the diagnosis of sleep apnea. J Clin Sleep Med. 2023 Jul 1;19(7):1337-1363. doi: 10.5664/jcsm.10532. PMID: 36856067; PMCID: PMC10315608.

• The discussion and conclusion can be made stronger by mentioning that this AI augmented WVU device can help in development of clinical prediction models that rely on accurate oxygen desaturation index (ODI) especially in cardiac patients planning to have cardiac procedures such as CABG or other procedure. 

Response: We appreciate the excellent suggestion by the reviewer and have incorporated the above sentence in the discussion and conclusion section of the updated manuscript.

SUGGESTIONS:

• The discussion section could further explore how the WVU device might be integrated into clinical care pathways for sleep apnea. What are the potential benefits and challenges of implementing this device in practice?

Response: Taking Reviewers comment into consideration, we now have included the following sentence along with a reference in the discussion section (page # 8).

AI augmented WVU device can also help in development of clinical prediction models that rely on accurate oxygen desaturation index (ODI) especially in patients planning to have surgical procedures such as CABG, bariatric surgery or arthroplasty among others. The device may help with better triaging of patients and reducing wait time for surgeries.

• Commenting on cost-effectiveness: If data are available, the authors could discuss the potential cost-effectiveness of the WVU device compared to traditional HSAT. 

Response: Since the technology has not been commercialized, we are unable to comment on the final cost of the device. However, based on our preliminary analysis on the cost of goods the basic model has the potential to be offered at a very reasonable price (lower than some of the healthcare monitoring gadgets available in the market).

• This is an important consideration for adoption of new technology.

Response: We agree with the reviewers comments.

• Expanding the limitations section: The authors could mention any potential limitations related to the AI algorithm, such as the need for external validation on other datasets or the potential for bias if the training data were not representative.

Response: The following sentences have been added to the limitations of the discussion section (page # 9) of the updated manuscript to address the reviewers’ comments:

“Furthermore, machine learning algorithms have their own limitations including need for external validation on external data and potential for bias if the training data is not representative of the patient population to be studied. The latter has the risk of widening health disparities in ethnic minorities along with introducing regional bias.”

• Discussing future directions

Response: Future research should focus on validating the device in a larger, more diverse population, along with evaluating its impact on patient outcomes.

• Enhancing figures: While the figures are clear, some additional annotations or labels could help guide the reader's interpretation.

Response: As suggested by the reviewer, we have clarified the figures by additional annotations.

Addressed

“This work is supported in part by funds from the National Science Foundation (NSF: Award # 2125872)”

Response: In addition to the funds from NSF the study was supported by the Department of Medicine, West Virginia University.

Response: Thank you for the suggestion. We now have added this sentence to the manuscript.

Response: Dr. Sharma is a full-time professor and faculty at the West Virginia University and receives salary from WVU. No other reimbursements were received for the study by any of the authors.

Response: As recommended this sentence has been added.

Response: Cover letter has been revised to include amended statements

3. In the online submission form, you indicated that your data will be submitted to a repository upon acceptance. We strongly recommend all authors deposit their data before acceptance, as the process can be lengthy and hold up publication timelines. Please note that, though access restrictions are acceptable now, your entire minimal dataset will need to be made freely accessible if your manuscript is accepted for publication. This policy applies to all data except where public deposition would breach compliance with the protocol approved by your research ethics board. If you are unable to adhere to our open data policy, please kindly revise your statement to explain your reasoning and we will seek the editor's input on an exemption.

Response: We appreciate the reminder and the importance of timely data deposition. Taking the Reviewers suggestion and comments, we now have uploaded an excel file with the data used for the analysis of this paper. 

In-text citations have been matched

The supporting document is called dataset

Reference list reviewed and rearranged to accommodate additional references

Reviewer #1: Data Access: the authors have not made all data fully available without restriction. In the Data Availability statement, the authors state:

"Data are available from the WVU Institutional Data Access / Ethics Committee (contact via email) for researchers who meet the criteria for access to confidential data"

This indicates there are some restrictions on openly sharing the underlying data. Researchers can request access through the WVU Institutional Ethics Committee, but would need to meet certain criteria to obtain the confidential data.

Dataset excel file now made available 

Minor Refinements:

Expanding the introduction: The authors could provide a bit more context on the current limitations of home sleep apnea testing (HSAT) devices and the potential advantages of the WVU device. This would help highlight the significance of their work.

Response: As suggested by the esteemed reviewer we have added the following sentence: “ Home sleep apnea testing (HSAT) on the other hand has high failure rates, cumbersome for patients and only captures single night data only. (lux etal)

Providing more details on the WVU device: The methods section could include some additional technical specifications about the WVU device, such as the specific sensors used, sampling rates, or key components of the AI algorithm. This would allow readers to better understand and potentially replicate the technology.

Response: Thank you for the suggestion to include additional technical specifications about the WVU device in the methods section. While we acknowledge the potential value of providing such details for readers' understanding and potential replication of the technology, we must also consider limitations related to patented information. As certain components of the WVU device are proprietary and protected by patents, we are restricted in disclosing specific details that could compromise intellectual property rights. However, we strived to provide as much relevant information as possible within the boundaries of what can be shared publicly, ensuring transparency while respecting confidentiality agreements and intellectual property protections.

Reporting confidence intervals: In addition to the point estimates for accuracy, sensitivity, specificity, etc., the authors could report 95% confidence intervals. This would provide a measure of the precision of these estimates.

Response: We thank the Reviewer for pointing this out. We have updated the Figure 4 to provide the 95% confidence intervals for AUC, sensitivity, and specificity values within the ROC plots for each of the AHI Thresholds 5, 15 and 30 events per hour. 

Discussing implications for clinical practice: The discussion section could further explore how the WVU device might be integrated into clinical care pathways for sleep apnea. What are the potential benefits and challenges of implementing this device in practice?

Response: We have added the following sentence to acknowledge the challenges- “potential challenges to incorporate WVU device are; limited comfort levels of physicians with this technology and the inertia of already established care pathways”.

Commenting on cost-effectiveness: If data are available, the authors could discuss the potential cost-effectiveness of the WVU device compared to traditional HSAT. This is an important consideration for adoption of new technology.

Response: Since the technology has not been commercialized, we are unable to comment on the final cost of the device. However, based on our preliminary analysis on the cost of goods the basic model has the potential to be offered at a very reasonable price (lower than some of the healthcare monitoring gadgets available in the market) 

Expanding the limitations section: The authors could mention any potential limitations related to the AI algorithm, such as the need for external validation on other datasets or the potential for bias if the training data were not representative.

Response: As suggested by the reviewers the following sentence has been added:

“Furthermore, machine learning algorithms have their own limitations including need for external validation on external data and potential for bias if the training data is not representative of the patient population to be studied. The latter has the risk of widening health disparities in ethnic minorities along with introducing regional bias.”

Discussing future directions: The conclusion could outline some key next steps for research, such as validating the device in a larger, more diverse population or conducting studies on the impact of the device on patient outcomes.

Response: The following sentence has been added to the discussion section: 

“Future research should focus on validating the device in a larger, more diverse population, along with evaluating its impact on patient outcomes.”

Enhancing figures: While the figures are clear, some additional annotations or labels could help guide the reader's interpretation. For example, labeling the sensitivity and specificity directly on the ROC curves.

Response: Reviewers comment and suggestion is appreciated. 

Taking reviewers comments into consideration we have expanded the figure legends to provide further details (see Figures 1 and 4). Figure 4 has been modified to provide 3 separate curves for each of the AHI thresholds 5, 15, and 30 events per hour along with the sensitivity and specificity as insets as part of the ROC curves. 

Reviewer #2: Dear authors, thanks for bringing this research to show the importance of new device capable of measuring oxygen level in people with different skin colors with fair and comparable accuracy with FDA approved device. It would be great to mention the name and model of FDA approved device. The introduction can be made stronger by mentioning recent research that AI can fill the gap of underdiagnosis of sleep disorders. The discussion and conclusion can be made stronger by mentioning that this AI augmented WVU device can help in development of clinical prediction models that rely on accurate oxygen desaturation index (ODI) especially in cardiac patients planning to have cardiac procedures such as CABG or other procedure. 

Response: We truly appreciate the kind remarks by our esteemed reviewer. Both the introduction (page # 3) and discussion (page # 8) sections have been bolstered with the suggestions provided by the reviewers. 

There are minor typo errors such as full stop after PPG at one place. The date format in methods can be modified to US standard. Otherwise, well written article!

Response: We appreciate the reviewer pointing out to the minor editing mistakes. We apologize for the oversight and have fixed it. We also truly appreciate the complements by our esteemed reviewer.

---

## [Editor Report · Decision Letter 1]

19 Apr 2024

Artificial Intelligence Augmented Home Sleep Apnea Testing Device Study (AISAP study)

PONE-D-24-04849R1

Dear Dr. Sharma,

We’re pleased to inform you that your manuscript has been judged scientifically suitable for publication and will be formally accepted for publication once it meets all outstanding technical requirements.

Kind regards,

Harpreet Singh Grewal

Academic Editor

PLOS ONE
---

## [Editor Report · Acceptance letter]

7 May 2024

PONE-D-24-04849R1 

PLOS ONE

Dear Dr. Sharma, 

I'm pleased to inform you that your manuscript has been deemed suitable for publication in PLOS ONE. Congratulations! Your manuscript is now being handed over to our production team.

Kind regards, 

on behalf of

Dr. Harpreet Singh Grewal 

Academic Editor

PLOS ONE